# Ordinary Gasoline Emissions Induce a Toxic Response in Bronchial Cells Grown at Air-Liquid Interface

**DOI:** 10.3390/ijms22010079

**Published:** 2020-12-23

**Authors:** Tereza Cervena, Michal Vojtisek-Lom, Kristyna Vrbova, Antonin Ambroz, Zuzana Novakova, Fatima Elzeinova, Michal Sima, Vit Beranek, Martin Pechout, David Macoun, Jiri Klema, Andrea Rossnerova, Miroslav Ciganek, Jan Topinka, Pavel Rossner

**Affiliations:** 1Department of Nanotoxicology and Molecular Epidemiology, Institute of Experimental Medicine of the CAS, Videnska 1083, 142 20 Prague, Czech Republic; tereza.cervena@iem.cas.cz (T.C.); kristyna.vrbova@iem.cas.cz (K.V.); antonin.ambroz@iem.cas.cz (A.A.); zuzana.novakova@iem.cas.cz (Z.N.); fatima.elzeinova@iem.cas.cz (F.E.); michal.sima@iem.cas.cz (M.S.); 2Department of Physiology, Faculty of Science, Charles University, Vinicna 7, 128 44 Prague, Czech Republic; 3Centre of Vehicles for Sustainable Mobility, Faculty of Mechanical Engineering, Czech Technical University in Prague, Technicka 4, 160 00 Prague, Czech Republic; michal.vojtisek@fs.cvut.cz (M.V.-L.); vit.beranek@fs.cvut.cz (V.B.); 4Department of Vehicles and Ground Transport, Czech University of Life Sciences in Prague, Kamycka 129, 165 21 Prague, Czech Republic; pechout@tf.czu.cz (M.P.); macound@tf.czu.cz (D.M.); 5Department of Computer Science, Czech Technical University in Prague, 121 35 Prague, Czech Republic; klema@fel.cvut.cz; 6Department of Genetic Toxicology and Epigenetics, Institute of Experimental Medicine of the CAS, Videnska 1083, 142 20 Prague, Czech Republic; andrea.rossnerova@iem.cas.cz (A.R.); jan.topinka@iem.cas.cz (J.T.); 7Department of Chemistry and Toxicology, Veterinary Research Institute, 621 00 Brno, Czech Republic; ciganek@vri.cz

**Keywords:** gasoline emissions, toxicity, air-liquid interface, MucilAir™, bronchial epithelial cells

## Abstract

Gasoline engine emissions have been classified as possibly carcinogenic to humans and represent a significant health risk. In this study, we used MucilAir™, a three-dimensional (3D) model of the human airway, and BEAS-2B, cells originating from the human bronchial epithelium, grown at the air-liquid interface to assess the toxicity of ordinary gasoline exhaust produced by a direct injection spark ignition engine. The transepithelial electrical resistance (TEER), production of mucin, and lactate dehydrogenase (LDH) and adenylate kinase (AK) activities were analyzed after one day and five days of exposure. The induction of double-stranded DNA breaks was measured by the detection of histone H2AX phosphorylation. Next-generation sequencing was used to analyze the modulation of expression of the relevant 370 genes. The exposure to gasoline emissions affected the integrity, as well as LDH and AK leakage in the 3D model, particularly after longer exposure periods. Mucin production was mostly decreased with the exception of longer BEAS-2B treatment, for which a significant increase was detected. DNA damage was detected after five days of exposure in the 3D model, but not in BEAS-2B cells. The expression of *CYP1A1* and *GSTA3* was modulated in MucilAir™ tissues after 5 days of treatment. In BEAS-2B cells, the expression of 39 mRNAs was affected after short exposure, most of them were upregulated. The five days of exposure modulated the expression of 11 genes in this cell line. In conclusion, the ordinary gasoline emissions induced a toxic response in MucilAir™. In BEAS-2B cells, the biological response was less pronounced, mostly limited to gene expression changes.

## 1. Introduction

Air pollution, notably road traffic pollution, is one of the most discussed topics related to health risks in large cities. It increases the incidence of cardiopulmonary and neurodegenerative disorders, influences reproduction, and is also a cancer risk [1,2,3]. As diesel engine exhaust was classified as a human carcinogen [4] and gasoline engine exhaust is possibly carcinogenic to humans [5], many cities have banned entry for diesel cars to city-centers [6], and electric vehicles are replacing combustion engine-powered cars. These decisions influence air quality, particularly in cities and other densely populated areas. Therefore, a thorough evaluation of the impact of these measures is necessary.

In terms of chemical composition, an engine exhaust represents a complex mixture of gases and particulate matter (PM), e.g., carbon mono- and dioxide, nitrogen oxides (NOx), volatile organic compounds, polycyclic aromatic hydrocarbons (PAHs), and heavy metals [7,8]. The individual component ratio is determined not only by the type of combustion engine, but also by the fuel (gasoline-ethanol blends) and how the engine is operated [9,10]. A recent study by Yang et al. compared four types of gasoline fuel with different blends of ethanol. Interestingly, total hydrocarbon, non-methane hydrocarbon, carbon monoxide, particulate emissions, and gaseous toxins were reduced for higher ethanol blends, but formaldehyde and acetaldehyde emissions substantially increased [11]. Due to the health risks associated with traffic-related air pollutants [12], the toxicity of these mixtures should be tested in conditions that realistically mimic human exposure.

We recognize four main branches of air pollution toxicology: epidemiological studies, controlled human studies, animal experiments, and in vitro-based tests. Following the concept of 3R (replacement, reduction, and refinement), new in vitro exposure systems and models emerged, enabling studies more relevant to human exposure. One of the most widely used models is three-dimensional (3D) cell cultures in different forms: cell cocultures, organoids, or reconstituent primary tissue models [13,14,15]. These 3D models mimic inter- and intracellular interactions found in in vivo conditions, and thus, help us understand the importance of a mucus barrier [16], mucociliary clearance and tight junctions [17,18], transcellular transport [19,20], and the presence of immune cells [21]. In terms of inhalation toxicology, several models originating from human lung tissue are available [22,23,24].

Traffic-related toxicity can be studied using various approaches, e.g., the exposure of cell cultures to complete emissions at the air-liquid interface (ALI) [25,26,27,28], exposure to suspended particles [29,30], or exposure to extracts obtained from particles collected on a filter. One of the advantages of particle extracts is the possibility to use them in traditional submerged conditions, but their chemical properties may vary depending on the solvents used for their preparation [31,32,33,34]. The tests based on the distribution of complete emissions represent the most realistic, real-world conditions; however, their application is associated with some limitations. They cannot be used for submerged cell cultures, and exposure systems have to deal with high particle loss before the sample reaches the cells, as well as with limited particle deposition [9,35]. In addition, growing the cells at the air-liquid interface may have negative impacts on cell cultures. Consequently, most of the studies did not exceed several hours of exposure [26,36,37,38], although some research groups were even able to maintain the ALI exposure for several weeks [39,40,41,42,43,44]. The fluctuation of air-pollutant concentration during the day should be also considered, and testing scenarios should ideally mimic these changes [45,46]. Complete emissions consisting of aerosol mixtures can cause systemic effects, such as the recruitment of inflammatory cells and cardiovascular toxicity along with local respiratory system responses such as cytokine production, oxidative stress, and cell death [47,48,49].

The airflow, temperature, humidity, and CO_2_ content have to be adjusted and monitored during the exposure to keep optimal conditions for cell cultures. In a previously published paper, we introduced an in-house exposure system for real-time exposure to complete emissions. This system allows repeated acute and long-term exposure to complete exhaust fumes from an engine [35].

In this follow-up research, we conducted a set of tests using commonly available gasoline (BA-95N, Čepro, 4.9% ethanol, 0.3% ETBE, referred to as E5), combusted in a typical automotive direct injection spark ignition (DISI) engine. We used BEAS-2B cells, a standard monolayer culture, and MucilAir™, a 3D model of human bronchial epithelial tissue [22,50]. The experiments, in which the cells were exposed to complete engine emissions for up to 5 days, were performed at ALI in a custom-made exposure chamber [35]. The biological response was evaluated based on transepithelial electrical resistance (TEER), mucin production and cytotoxicity measurements, DNA strand breaks detection, and gene expression analysis.

## 2. Results

### 2.1. TEER Measurement, Cytotoxicity Assays, and Mucin Production Quantification

TEER values were measured before and after exposure to complete emissions/control air, as described in Section 4.3 and Section 4.4. Although BEAS-2B cells were able to form a uniform monolayer without medium leakage, we were not able to measure any change of TEER values (200 ± 13 Ω·cm^2^). This finding indicates a low number of tight junctions and a lack of ability to form a polarized cell monolayer. In contrast, the TEER measurement was an important indicator of cell model integrity in MucilAir™ (Figure 1). After one-day exposure, we observed a significant (*p* ≤ 0.05) drop, but no difference between the exposed and control cells (Figure 1A). All values were well above the minimal acceptable value provided by the manufacturer (200 Ω·cm^2^). The five-day exposure led to a slight decrease in TEER values over time in the exposed samples (Figure 1B). The decrease was significant at T3–T5 when compared with the controls. At T5, a significantly lower TEER value than in the exposed samples at T0 was observed. Despite the decrease, all TEER values were well above the minimal acceptable value. In contrast, the control TEER values were comparable at T0–T4; at T5 they were significantly higher than at T0. In general, MucilAir™ exposure to complete emissions had a relatively weak, but significant effect on cell model integrity.

Cytotoxicity was measured in the control and exposed samples at each time point in the cell culture medium as LDH and AK activity (Figure 2 and Figure 3). The results had mostly similar trends, but the response was more pronounced for LDH (Figure 2A–D). While no effect was observed in the MucilAir™ after one-day exposure (Figure 2A), we detected a slight, although statistically significant increase in LDH leakage at T3 and T4 in the exposed samples in the five-day exposure experiment. At T5, no significant difference between the control and exposed samples was detected, probably reflecting fluctuations in the biological response of the exposure system (Figure 2B). BEAS-2B cell samples showed a significant (*p* ≤ 0.05) increase in LDH leakage after one-day exposure for both the exposed and control samples (Figure 2C). The five-day exposure of BEAS-2B cells led to an almost linear increase of LDH leakage in both sets of samples (Figure 2D). This change was significant (*p* ≤ 0.05) at T2–T5 and at T1–T5, for the exposed and control samples, respectively. At T4–T5, this increase exceeded 50% cytotoxicity for the control samples. Thus, incubation of BEAS-2B cells in the exposure system, rather than the effect of the complete emissions, was probably a major factor affecting LDH leakage in this cell line. AK activity in the culture medium was lower overall (Figure 3A–D). One-day exposure of MucilAir™ led to a decrease of AK activity for both sets of samples, although the difference was significant (*p* ≤ 0.05) for the exposed samples only (Figure 3A). The five-day exposure of MucilAir™ had a similar trend of AK activity as LDH leakage in the same samples (Figure 2B and Figure 3B). Cytotoxicity in the exposed samples was significantly (*p* ≤ 0.05) higher at T0 and T2–T5, while in the control samples AK activity was minimal. In BEAS-2B cells, AK activity was low (less than 10%) and increased (*p* ≤ 0.05) at T3–T5 and T4–T5, for the exposed and control samples, respectively. We observed a significantly higher (*p* ≤ 0.05) AK activity in the exposed samples when compared with the controls at T1–T5 (Figure 3D).

Mucin production quantification was done before and after exposure to complete emissions, as described in Section 4.3. One-day exposure of MucilAir™ (Figure 4A) led to a decrease in mucin production in both the exposed and control samples (*p* ≤ 0.05). A significant decrease was also observed for the exposed, but not control samples after five days of exposure at T1–T5 (*p* ≤ 0.05) (Figure 4B). Mucin production in BEAS-2B cells was approximately 3-fold lower compared to the MucilAir™. One-day exposure showed a significant (*p* ≤ 0.05) decrease in both the control and exposed cells (Figure 4C). The time-dependent increase was detected at five days of exposure for the exposed and control cells (*p* ≤ 0.05 for T3–T5) (Figure 4D).

### 2.2. DNA Breaks Detection

Phosphorylation of histone H2AX was analyzed in cell lysates, prepared from cells collected at time points T1 and T5 to detect double-stranded DNA breaks. Although one-day exposure did not induce phosphorylation of histone H2AX in MucilAir™, a significant increase of DNA damage was observed after five days when compared with one-day exposure (Figure 5A). In addition, a significantly elevated (*p* ≤ 0.05) histone H2AX phosphorylation in the exposed compared to the control samples was found after five days of exposure. In BEAS-2B cells, exposure to complete emissions did not induce any change after one day of exposure. It was interesting to note that the exposed cells had a significantly (*p* ≤ 0.05) lower amount of phosphorylated histone H2AX after five days, in comparison with one-day exposure (Figure 5B). We also observed a significant (*p* ≤ 0.05) difference between the exposed and control samples after five days of exposure.

### 2.3. mRNA Expression Analysis

We first compared the effect of complete emissions on mRNA expression between the tested models and the control samples after one-day and five-day exposure. In the MucilAir™ system, the response was generally very weak. No deregulated mRNA was detected after one-day treatment; five-day exposure increased the expression of *GSTA3* (glutathione S-transferase alpha 3) and decreased the expression of *CYP1A1* (cytochrome P450 family 1 subfamily A member 1) (Table 1).

In contrast, in BEAS-2B cells, we detected 39 deregulated mRNA after one-day exposure and 11 deregulated mRNA after five-day exposure. Interestingly, after short exposure, the expression of most of the genes was upregulated. The notably affected processes included: apoptosis (*BID*, BH3 interacting domain death agonist, *CASP1*, caspase 1; *TNFSF10*, TNF superfamily member 10), DNA damage and repair (e.g., *PARP1*, poly(ADP-ribose) polymerase 1; *PCNA*, proliferating cell nuclear antigen), endoplasmic reticulum stress (e.g., *HERPUD1*, homocysteine inducible ER protein with ubiquitin-like domain 1; *SERP1*, stress associated endoplasmic reticulum protein 1; *SYVN1*, synoviolin 1), heat shock response (genes encoding various families of heat shock proteins, including, e.g., *HSPA1A*, *HSPA8*, *HSPA9*, *HSPD1*, and *HSPE1*), mitochondrial energy metabolism (*MDH1*, *MDH2*; malate dehydrogenase 1, 2; *SDHD*, succinate dehydrogenase complex subunit D; *SUCLA2*, succinate-CoA ligase ADP-forming subunit beta), and oxidative stress (*GPX1*, glutathione peroxidase 1; *SOD1*, superoxide dismutase 1). After five days of exposure, the expression of eight genes was downregulated, including those involved in apoptosis (*BIRC3*, baculoviral IAP repeat-containing 3; *MCL1*, MCL1 apoptosis regulator), oxidative stress (*NUDT15*, nudix hydrolase 15), and immunotoxicity (*IL6*, interleukin 6; *IL1A*, interleukin 1 alpha; *PTGS2*, prostaglandin-endoperoxide synthase 2) (Table 2). We found no common genes for these exposure conditions.

As the response to complete emissions exposure significantly differs between MucilAir™ and BEAS-2B cells, we further aimed to identify mRNA differentially expressed in both models after individual treatment periods. We detected 198 and 172 differentially expressed genes for a comparison between model systems after one-day and five-day treatment, respectively (Appendix A). These genes were involved in all investigated biological pathways. For the short exposure period, the number of mRNA downregulated in the MucilAir™ tissues, when compared with BEAS-2B cells, was similar to the number of upregulated genes. In contrast, after five days of exposure, the downregulated mRNAs were slightly more common than those upregulated (101 vs. 97 and 68 vs. 104, for upregulated and downregulated mRNA after one-day vs. five-day exposure, respectively).

Further investigation revealed that there were 58 and 32 unique differentially expressed mRNA for a comparison between models after one-day and five-day exposure, respectively. We also found 140 common genes whose expression differed between MucilAir™ and BEAS-2B cells, regardless of the exposure period (Figure 6; Appendix A). We observed a very similar response for the control samples (data not shown). These results suggest that the type of model system, rather than the exposure conditions, is a key factor determining the biological response in vitro at the air-liquid interface after treatment with complete emissions.

To gain more information on the biological impacts of complete emissions exposure and differences between model systems, we performed pathways identification using the functional enrichment analysis (Table 3). Treatment of BEAS-2B cells for one day resulted in over-representation of Citrate cycle and Mismatch repair pathways when compared with the controls, while the longer, five-day exposure was associated with enrichment of TNF signaling pathway and Pathways of cancer. In a comparison of model systems, significant over-representation of pathways linked with fatty acid and carbon metabolisms, as well as with apoptosis, beta-oxidation, and p53 signaling, was observed in MucilAir™ when compared with BEAS-2B cells treated for one day and five days, respectively. The common differences between the model systems in response to complete emissions exposure were associated with enrichment of Apoptosis, Citrate cycle, and Drug metabolism—cytochrome P450 pathways (Table 3).

Selected mRNA expression data from RNA sequencing were verified by quantitative real-time PCR (qRT-PCR). Specifically, we chose representative genes from those exposure conditions for which significant differences between exposed and control samples were found (Table 1 and Table 2). These genes included *CYP1A1* (MucilAir™, five-day exposure), *BID*, *PCNA* (BEAS-2B cells, one-day exposure), *IL6*, and *PTGS2* (BEAS-2B cells, five-day exposure) (Appendix A). We detected a significant correlation between RNA sequencing and qRT-PCR data across all the transcripts (Pearson R = 0.93, *p* < 0.005). The correlation was calculated based on log2FC values obtained for individual transcripts using the respective method. Significant results were detected for three out of five verified genes.

## 3. Discussion

In this study, we investigated the toxicity and biological response of two cell models grown at the air-liquid interface, after their exposure to complete emissions generated by direct injection spark ignition engine powered by ordinary gasoline, containing 4.9% (*v*/*v*) ethanol. The results suggest that in MucilAir™, the exposure induces a cellular response manifested by increased levels of cytotoxicity markers (LDH, AK), and a disturbance of parameters that characterize the overall conditions of the cells (TEER, mucin) and the DNA damage (H2AX phosphorylation). Changes of the regulation of biological processes (mRNA expression) were weak in this model system. The response in BEAS-2B cells was generally more pronounced. However, we assume that it was probably induced by cultivation conditions in the exposure system (the impact of airflow) rather than by the effects of emissions. We previously conducted a study that had a similar design, but tested the fuel containing 20% (*v*/*v*) ethanol (E20). In contrast to the current study, we concluded that the biological effects of E20 fuel were minimal [51].

The TEER measurement and mucin production were used as parameters evaluating the overall conditions of cell cultures. TEER that reflects tissue integrity and tight junctions [52] significantly decreased at T1 in both the exposed and control samples of MucilAir™. Similarly, at other intervals, a trend to lower TEER was observed in the exposed cultures, although the changes were mostly not significant when compared with T0. However, a significantly decreased TEER in the exposed cells was noted at T5. In contrast, in the control samples, TEER was significantly higher at T5 when compared with T0. Importantly, at T3–T5, TEER in the exposed samples decreased in comparison with the controls, suggesting toxic effects of complete emissions in this model system. It should be emphasized that a decrease of TEER values at all time points was relatively minor and tight junctions were most likely not significantly affected, as TEER remained well above the 200 Ω·cm^2^ limit. In comparison with E5, the results obtained for E20 in our previous study showed a significant increase of TEER values at T2–T5 time points when compared with T0, suggesting a lower toxicity of fuel with higher ethanol content [51].

Mucin is produced in the airways as a mechanism of protecting the organism against negative environmental factors [16]. We have reported the induction of mucin production in BEAS-2B cells, but not in MucilAir™ exposed to E20 emissions [51]. Although E5 emissions exposure was also associated with increased mucin production in BEAS-2B cells, a significant decrease was detected for MucilAir™ samples. It should be further noted that for the samples exposed to control air, the decrease was less pronounced and the overall values did not significantly differ from those at T0. This finding suggests a compromised protection of MucilAir™ against the deleterious impacts of E5 emissions, and may be related to the overall negative effects of E5 emissions that are manifested by the interference with various cellular functions, including protein synthesis. Such effects were observed in various cell models, including BEAS-2B cells, after exposure to particulate matter and organic extracts from PM [53,54].

In agreement with the TEER and mucin data, the detection of extracellular activities of LDH and AK revealed a cellular response after E5 exhaust exposure. These effects tended to increase with exposure time and were more pronounced in BEAS-2B cells and for the LDH activity. In MucilAir™ samples, the highest LDH cytotoxicity values were around 6%, while for BEAS-2B cells they exceeded 40%. These results indicate that the impact of E5 emissions exposure in MucilAir™, albeit significant, has limited biological significance. In contrast, for BEAS-2B cells this exposure is cytotoxic, even after a relatively short exposure period of two days. These data further differentiate E5 fuel from E20, for which the cytotoxic effects were not detectable, with the exception of LDH activity in BEAS-2B cells, again suggesting the lower toxicity of this fuel [51]. It should be further mentioned that in BEAS-2B cells, a significant increase of LDH leakage was detected even in the control samples, particularly at T3 and T4, when the LDH activity significantly exceeded the values for the exposed cell cultures. Thus, in this cell line the toxic response reflects not only the effect of emissions but also the reaction of the cell cultures to cultivation conditions in the exposure system, probably due to airflow. It has been previously reported that airflow may negatively affect the viability of alveolar cells [55,56].

Histone H2AX phosphorylation is commonly used as a marker of double-stranded DNA breaks [57]. Its assessment has been used to analyze the genotoxicity of various air pollutants, including, e.g., polycyclic aromatic hydrocarbons, their derivatives, and other components of diesel exhaust [58,59,60,61]. However, to date the effect of complete gasoline emissions on histone H2AX phosphorylation has only been investigated in our recent study [51], in which we found the induction of DNA damage in BEAS-2B cells exposed to E20 fuel, but no effect in the MucilAir™ model. This is in contrast with our present report, in which we detected significant DNA damage in the 3D model, but decreased levels of gamma-H2AX in BEAS-2B cells exposed for 5 days to E5 emissions. In addition, in BEAS-2B cells, DNA damage in the controls was higher than in the exposed samples. Although unexpected, these results may be explained by the higher toxicity of E5 emissions, negatively impacting cellular processes. A similar phenomenon was reported by Yamamori et al. for human lung carcinoma cells A549 [62]. The authors argue that a DNA double-strand break repair was suppressed due to the induction of endoplasmic reticulum (ER) stress, caused by the intratumoral environment. In our study, however, the process is probably caused by a different mechanism, as the expression of genes related to ER stress was not detected for a comparison of the exposed and control samples for either cell model. The conflicting results obtained for the test models in our study again highlight differences between the MucilAir™ model and BEAS-2B cells grown at the ALI.

For a comprehensive assessment of the biological effects induced by complete gasoline exhaust at the ALI, we analyzed the expression of 370 selected genes involved in the processes potentially linked with the toxicity of gaseous and particle pollutants present in emissions. As discussed in our previous study [51], a limited number of reports have investigated the impact of complete diesel and/or gasoline exhaust emissions on gene expression in various cellular systems in vitro. Bisig et al. analyzed the toxic effects of exhaust aerosols from ethanol-gasoline blends after 6-h exposure, followed by a 6-h post-incubation period in human bronchial cells (16HBE14o-), supplemented with human monocyte-derived dendritic cells and monocyte-derived macrophages. In the cells treated with ethanol-gasoline blend aerosols, the authors found no induction of the expression of studied genes including those related to oxidative stress (*HMOX1*, *SOD2*, *GSR*) and inflammation (*TNFα*, *IL-8*). Interestingly, the emissions from fuel containing 85% ethanol reduced the expression of *SOD2* and *IL-8*, although the changes were not significant [38]. In another study focused on the effects of complete exhaust from gasoline direct injection cars, which involved both a multi-cellular human lung model (16HBE14o-cell line, macrophages, and dendritic cells) and MucilAir™ tissues, no oxidative stress- (*HMOX1*, *SOD2*, *NQO1*) or inflammation-related (*CXCL8*, *TNFα*) genes were affected after a single 6-h exposure. However, repeated exposure resulted in the induction of *HMOX1* and *TNFα* genes in the multi-cellular model [37]. The combined effect of respirable volcanic ash and complete exhaust from a gasoline vehicle was studied in a multi-cellular model consisting of A549 alveolar type II-like cells, complemented with human blood monocyte-derived macrophages and dendritic cells, cultured at the ALI [63]. The cells were exposed to gasoline emissions for 6 h repeatedly and for 18 h to volcanic ash, but the authors did not observe any significant changes of expression of the genes associated with apoptosis (*FAS*, *CASP7*), oxidative stress (*HMOX*, *NQO1*), and pro-inflammatory markers (*IL8*, *IL1B*, *TNFA*). These results suggest that the biological effects of complete emissions from gasoline cars are relatively weak and require repeated exposure to induce gene expression changes.

In contrast to these studies, we extended the exposure periods to complete emissions to 5 days, with the aim to more realistically mimic real-life scenarios. This approach, along with the analysis of a higher number of genes, allowed us to better characterize the biological impacts of gasoline emissions. Similar to our previous study [51], the response of the MucilAir™ tissues was characterized by the expression changes of a low number of deregulated genes, although, in contrast to E5, E20 affected gene expression after a one-day exposure. The emissions from both fuels modulated the expression of *CYP1A1*, a gene encoding a protein involved in the metabolism of various xenobiotics, including polycyclic aromatic hydrocarbons (PAHs). We have previously detected the expression of this gene after MucilAir™ exposure to benzo[a]pyrene [64]. Although downregulation of *CYP1A1* expression observed in the present study might seem unexpected, it can be explained by the induction of an inflammatory response after emissions exposure [65]. The expression of *GSTA3* was upregulated, which is in line with a study that found this gene to be positively regulated by the exposure to PAHs present in organic extract from diesel exhaust particles [66].

Interestingly, the response of BEAS-2B cells induced by emissions from E5 and E20 significantly differed. While no gene expression modulation was observed after E20 treatment, E5 exposure was associated with the deregulation of 39 and 11 genes, after one-day and five-day exposure, respectively. It should be noted that the short exposure period was associated with the upregulation of most of the genes, including, e.g., those involved in apoptosis, DNA damage response, endoplasmic reticulum stress, heat shock, or oxidative stress. These processes are commonly induced in response to the exposure to toxic compounds. In contrast, the five-day treatment resulted in the downregulation of genes involved in apoptosis, oxidative stress or immune response. Although the effect was relatively weak, we may conclude that this exposure negatively affects biological changes in BEAS-2B cells that activate a protective cellular response. We have previously speculated that a lack of gene expression induction, observed after E20 exposure in BEAS-2B cells, may be associated with unfavorable incubation conditions [51]. The results obtained for E5 treatment, also underlined by our observations for general parameters of toxicity (particularly LDH leakage and mucin production), may reflect a similar phenomenon.

The observed higher toxicity of ordinary gasoline when compared with E20 fuel may be explained by differences in the chemical composition of emissions, as well as by the mass of PM generated by individual fuels. Although organic extracts from E5 PM contained a greater proportion of some PAHs than E20 PM, the total concentration of carcinogenic PAHs was about 74% higher in the extracts from E20 than E5 PM. In contrast, E5 combustion generated up to 3-fold more PM than the ethanol-containing fuel. This discrepancy may be responsible for the more pronounced biological effects of ordinary gasoline exhaust.

A comparison of gene expression between both model systems, induced after the short and long exposure period, revealed differences for about 50% of the studied genes. This result was also observed for E20 in our previous study [35], which highlights the crucial role of the cell model used in toxicological tests for the evaluation of toxicity and interpretation of the biological effects of the tested compounds.

We used functional enrichment analysis to obtain more data on pathway over-representation associated with exposure to complete emissions and differences between model systems. Although the analysis was limited by the number of deregulated genes detected by the targeted expression panel, we observed enrichment of pathways involved in carbon metabolism and DNA repair after one-day exposure, and an immune response as well as carcinogenesis after five days of treatment of BEAS-2B cells. The functional enrichment analysis further suggested that the differences between BEAS-2B cells and MucilAir™ in response to the treatment were related to over-representation of pathways associated with apoptosis and carbon and xenobiotics metabolism.

In summary, our study contributes to the understanding of the toxicity of complete emissions in the in vitro systems and provides new knowledge on the long-term exposure of 3D and standard cell cultures incubated at the ALI. However, there are some limitations to our data, stemming mainly from the fact that a panel of pre-selected genes, rather than whole-genome approach, was used for the gene expression analysis. A more detailed assessment of gene expression changes would allow the potential detection of processes not covered by our custom assay. As the study design, determined by budget restrictions, did not allow us to perform whole-genome RNA sequencing, future studies are needed to fill these knowledge gaps. Finally, mRNA expression was not confirmed on protein levels, particularly because of a limited amount of biological material available for subsequent analyses after exposure to the complete emissions. Assessment of protein expression would help to explore mechanisms of toxic responses caused by the ordinary gasoline emissions in more details.

## 4. Materials and Methods

### 4.1. Cell Cultures

In this study, we used two types of cell models to investigate the complete emission effects: BEAS-2B, a human bronchial epithelial cell line (CRL-9609TM, ATCC^®^, Manassas, VA, USA), and 3D lung tissue model MucilAir™ (Epithelix Sàrl, Geneva, Switzerland). The BEAS-2B cells represent an immortalized adherent cell line derived from lung autopsy [50], while MucilAir™ represents a fully differentiated bronchial epithelium reconstituted from primary human cells [22]. It consists of human basal, goblet, and ciliated cells and, unlike BEAS-2B, which typically grows in submerged conditions, the MucilAir™ lung model is optimized for air-liquid interface by the manufacturer and grows on cell inserts. Its features include a lack of overgrowing after more extended cultivation periods, the ability to heal, as well as cell stratification, formation of tight junctions, mucus production, metabolic activity, and cilia beating. Our samples were obtained from a 64-year-old Caucasian female, non-smoker, with no pathology reported.

Both cell models were cultivated at air-liquid interface at standard conditions (37 °C, 5% CO_2_, relative humidity > 90%), and we aimed to keep the same parameters during exposure to emissions to avoid any discrepancy. Unlike MucilAir™, BEAS-2B cells must be adapted to ALI incubation before exposure. Using serum-free cultivation conditions (BEGMTMkit CC-3170; Lonza, Basel, Switzerland), we seeded 100,000 cells/insert in 24-well format Transwell^®^ cell culture inserts (Sigma-Aldrich, St. Louis, MO, USA). Twenty-four hours later, we removed the apical medium. The inserts were kept at ALI for another 24 h to form a monolayer with no medium leakage from the basal part, and were prepared for exposure.

3D tissue models MucilAir™ were monitored for three weeks before exposure to determine their stability. During this period, the culture medium (Epithelix Sàrl, Geneva, Switzerland) was changed every 2–3 days, and an apical wash was performed every week. Cell inserts were monitored for protrusions on the surface, perforations in its structure, and the presence of beating cilia under a light microscope (Olympus CKX41, Tokyo, Japan; 200× magnification).

### 4.2. The Complete Emissions Exposure System

Exposure to complete emissions was conducted using a transient engine dynamometer on a Euro 5 direct injection spark ignition engine, as described in [35,51]. Briefly, in this study, the engine was operated at the speed and load points of the same model engine in a typical European middle-class passenger car (Škoda Octavia) during a World Harmonized Light Vehicle Test Cycle (WLTC) driven on a chassis dynamometer. The engine was run on commonly available gasoline (E5; BA-95N, Čepro, 4.9% ethanol, 0.3% ETBE).

Raw exhaust gases were diluted with filtered air at a constant dilution ratio of 10:1 and the diluted exhaust was simultaneously sampled onto 70-mm diameter fluorocarbon-coated glass filters (PallFlex, Pall, Portsmouth, UK) for later particulate matter extraction and into an exposure box for direct real-time cell exposure [35]. To ensure proper incubation conditions in exposure box, the exhaust sample (and in a parallel branch, filtered air as control) was enriched to 5% CO_2_ and the sample was humidified by a selective membrane humidifier (Nafion, model no. FC125-240-5MR, PermaPure, Lakewood, NJ, USA) to over 85% relative humidity, heated to 37 °C, and distributed among symmetrical channels among the cell inserts (25 cm^3^/min/insert) to allow unassisted deposition by diffusion on cell cultures. Exposure and technical details are described in [35].

Details on particle concentrations and filter loading for diluted and undiluted exhaust are provided in Table 4. The mean concentration of particulate matter in the diluted exhaust, as determined by gravimetric analysis, was 0.175 mg/m^3^ (0.201 ± 0.089 mg/m^3^ for cold start tests and 0.148 ± 0.050 mg/m^3^ for warm start tests). The mean concentration of black soot in undiluted exhaust, as measured by a photo-acoustic analyzer (AVL Microsoot Sensor, AVL List GmbH, Graz, Austria), was 0.3 mg/m^3^. The mean mass concentration of particles as determined from size distributions based on electric mobility (measured by Engine Exhaust Particle Sizer, TSI, in diluted exhaust, corrected for dilution) and aerodynamic diameter (measured by Electrostatic Low Pressure Impactor, Dekati, high-temperature version running at 160 °C, in undiluted exhaust) size distributions were 0.6–0.7 mg/m^3^ in undiluted exhaust. The mean particle concentrations in raw exhaust were 5–8 million particles per cm^3^ (#/cm^3^) for cold start and around 2 million #/cm^3^ for a warm start test, corresponding to approximately 5–8 × 10^5^ #/cm^3^ for cold start and 2 × 10^5^ #/cm^3^ for warm start tests at the exposure chamber inlet. Some particles, about one-third, were lost in the diffusion membrane humidifier; a low flow through the humidifier and high humidity of the outgoing sample made it impossible to avoid losses [35]. It is estimated that most of the losses can be attributed to the diffusion of very small particles and that the losses on the mass basis are relatively small. It is also estimated that the deposition rate of particles by diffusion is about 2% [35]. At 0.175 mg/m^3^ average particle mass concentrations in the diluted exhaust, 25 cm^3^/min flow rate per insert, four 30-min tests per day for five days, a total of 15 dm^3^ was passed into each insert, containing 2625 ng of particles. At an assumed 2% deposition rate, about 52.5 ng, or about 50 ng, of particulate matter was deposited in each insert after five-day exposure and about 10 ng after one-day exposure (Table 5).

For chemical analysis, particulate matter collected on filters during each exposure was extracted using dichloromethane/cyclohexane, pooled accordingly to the exposure scheme, and analyzed by HPL with fluorimetric detection. Detailed information on collected particulate matter (PM) and concentrations of polycyclic aromatic carbons in E5 is reported in Appendix A. For reference, results from our recent study, focused on the biological effects of fuel containing 20% ethanol [51], are also provided.

### 4.3. Exposure Scheme

The exposure scheme was previously described in detail in [35,51]. Briefly, we aimed for realistic human exposure to exhaust fumes, particularly with repeated exposure during the day. Highly dynamic operations and cold starts are responsible for a large portion of emissions to which humans are exposed, and these relatively short exposure periods may be repeated during the day, for example as a person commutes to/from work. The exposure was conducted in our in-house developed exposure system [35], and two exposure schemes were used (Figure 7). The exposure unit consisted of: two WLTC runs (once with a cold start and once with a hot start); a 2-h block when the engine was actively cooled (control and treated cells were exposed to filtered ambient air; 5% CO_2_, over 85% relative humidity, and 37 °C); and two WLTC runs (once with a cold start and once with a hot start). This combination of conditions allowed us to expose the cells repeatedly for one and five days. The control samples were exposed to filtered ambient air (5% CO_2_, over 85% relative humidity, and 37 °C).

Prior to the exposure to complete emissions and control air (time point T0), we collected a medium and measured the TEER to obtain data from cells which were neither affected by handling nor exposure. Thereafter, the cells in portable exposure boxes were transported (37 °C, sealed to avoid the loss of a highly humid and CO_2_-enriched environment) to an exposure facility, where the boxes were connected to the exposure system, and exposure commenced. After exposure, the boxes were secured and transported back to the laboratory for: (a) one-day exposure (time point T1—TEER measurement, medium collection, cell collection); (b) five-day repeated exposure (time points T1–T4—TEER measurement, medium change and storage, cells placed to the incubator overnight); (c) five-day repeated exposure (time point T5—TEER measurement, medium collection, cell collection).

### 4.4. TEER Measurement and Mucin Production Quantification

TEER was used as a non-destructive quantitative method for measuring the cell culture integrity and ability to form tight junctions. The measurement was conducted using EVOM2 ohm meter (World Precision Instruments, Sarasota, FL, USA) with an STX2 electrode as previously described [51]. Each cell insert was measured three times at each time point.

Mucin production was quantified in a cell insert apical wash at each timepoint, using sandwich enzyme-linked lectin assay (ELLA) developed by Epithelix Sàrl (Geneva, Switzerland), with modifications reported in [51].

### 4.5. Cytotoxicity Assays

Cytotoxicity was measured as enzyme activity in a basal medium collected at each time point. For lactate dehydrogenase measurement, the Cytotoxicity Detection Kit (Roche, Basel, Switzerland) was used; adenylate kinase activity was detected using the Adenylate Kinase Cytotoxicity Assay Kit (Abcam, Cambridge, UK). The assays were performed according to the manufacturer’s instructions, and the results are presented as the percentage of cytotoxicity relative to the positive control (1% *v*/*v* Triton X-100, 1 h, and 37 °C).

### 4.6. Phosphorylation of Histone H2AX

At the T1 time point for one-day exposure and T5 time point for five-day exposure, double-stranded DNA breaks were measured. The detection of serine 139 phosphorylation of histone H2AX (ɣ-H2AX) was assessed in cell lysates using the ELISA kit (HT ɣ-H2AX Pharmacodynamic Assay; Trevigen, Gaithersburg, MD, USA) according to the manufacturer’s instructions. The level of H2AX phosphorylation is expressed in pM, based on the standard curve (ɣ-H2AX standard provided by the manufacturer). 

### 4.7. mRNA Expression Analysis

The expression of mRNA was assessed in samples collected at two time points: after one day and five days of exposure to complete emissions, as described in detail in our previous study [51]. The targeted mRNA expression analysis was performed using Human Molecular Toxicology Transcriptome panel (QIAseq Targeted RNA Panel, Qiagen, Hilden, Germany). This panel allowed us to detect the expression of 370 genes that play a role in apoptosis, necrosis, DNA damage and repair, mitochondrial energy metabolism, fatty acid metabolism, oxidative stress and antioxidant response, heat shock response, endoplasmic reticulum stress and unfolded protein response, cytochrome P450s and phase I drug metabolism, steatosis, cholestasis, phospholipidosis, and immunotoxicity (for a complete list, see Appendix A).

Total RNA was extracted using NucleoSpin RNA XS kit (Macherey-Nagel, Düren, Germany) according to the manufacturer’s instructions. The concentration of isolated RNA was measured with a Nanodrop ND-1000 spectrophotometer (Thermo Fisher Scientific, Wilmington, DE, USA). A total of 400 ng of RNA was used for library preparation according to the previously published protocol [51]. The library concentration was determined by Qubit 1x dsDNA HS Assay Kit on Qubit 4 fluorometer (Thermo Fisher Scientific, Wilmington, DE, USA). Libraries were validated on the Fragment analyzer system (Agilent Technologies, Santa Clara, CA, USA) using HS NGS Fragment Kit. For sequencing, the NextSeq^®^ 500/550 Mid Output Kit v2.5 (300 cycles) and the NextSeq system (Illumina, San Diego, CA, USA) were used.

RNA sequencing data were processed with NF-CORE RNASeq pipeline (https://github.com/nf-core/rnaseq, version 1.3) [67] as previously reported [51]. The reads were mapped to the reference genome GRCh38.p12. DESeq2 with default parameter settings was applied to normalize the read counts and to identify the differences in gene expression between the sample groups [68]. For multiple testing correction, the Benjamini & Hochberg method was used. To account for the gender differences of subjects from which the MucilAir™ tissues and BEAS-2B cells originated, all genes located on chromosome X were omitted in gene expression analyses. To perform functional enrichment analysis of significantly differentially expressed genes, ToppGene Suite (https://toppgene.cchmc.org/enrichment.jsp) was used. The calculations were performed with the feature “Pathways/BioSystems: KEGG” [69].

### 4.8. Quantitative Real-Time PCR Analysis

For synthesis of the cDNA, 400 ng of total RNA was mixed with components of the Transcriptor High Fidelity cDNA Synthesis Kit (Roche, Mannheim, Germany), including 2 µL of deoxynucleotide mix (10 mM), 1 µL anchored-oligo(dT)18 primer (50 pmol/mL), and 2 µL random hexamer primer (600 pmol/mL). The mixture was incubated at 55 °C, 30 min, and 85 °C, 5 min. The cDNA samples were diluted in nuclease-free water and tested using TATAA Probe^®^ GrandMaster Mix (TATAA Biocenter AB, Goteborg, Sweden). All reactions were run in duplicates. The qPCR analysis was performed in 10 μL reactions on LightCycler^®^480 II qPCR instrument (Roche, Mannheim, Germany). The reaction mixture contained 5 µL of TATAA Probe^®^ GrandMaster Mix, 2.5 µL of nuclease-free water, 0.5 µL of a respective TaqMan assay (Thermo Fisher Scientific, Waltham, MA USA) (Appendix A), and 2 µL of diluted cDNA. The cycling conditions were: incubation at 95 °C, 60 s, followed by 40 cycles of incubation at 95 °C, 5 s, and 60 °C, 30 s. Interplate calibrator was used in triplicates in each plate to compensate the possible differences between runs. The Ct values were used to calculate the log2 fold changes of gene expression using the delta delta Ct method. The expression of the target genes was normalized to the reference genes (ACTB, TOP1).

### 4.9. Statistical Analysis

The cell cultures were exposed in biological triplicate per parameter. Technical duplicates of each biological replicate were done for AK, LDH, and ɣ-H2AX. TEER values were obtained from a total of nine inserts per sample. The parameters were compared using two-way ANOVA with Sidak’s (post hoc) multiple comparison test and using Student’s *t*-test (GraphPad Prism version 8 (GraphPad Software Inc., San Diego, CA, USA)). Data were expressed as mean ± standard deviation (SD). Significance values ≤ 0.05 were considered significant. Venn diagrams were created in the Bioinformatics & Evolutionary Genomics tool (http://bioinformatics.psb.ugent.be/webtools/Venn/).

## Figures and Tables

**Figure 1 ijms-22-00079-f001:**
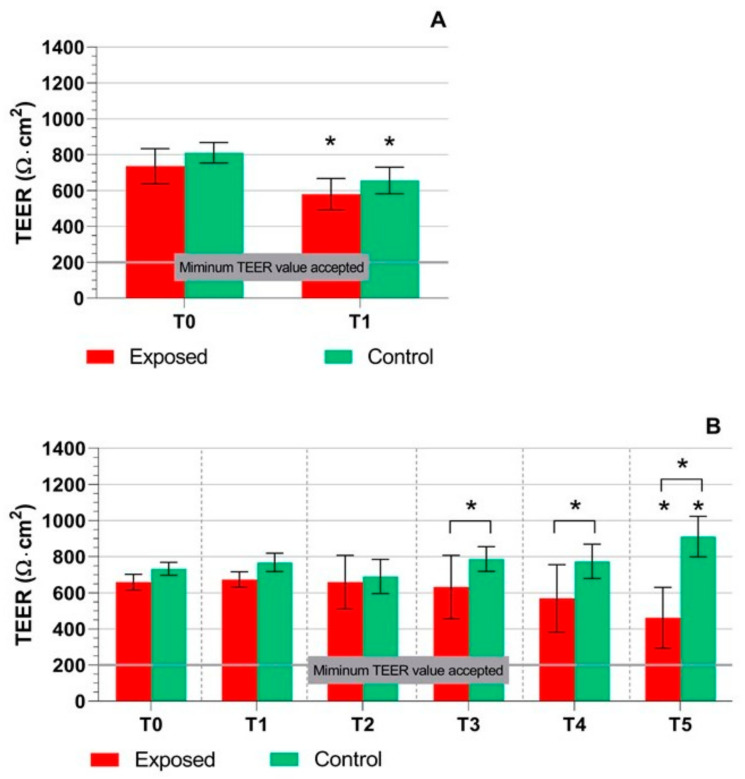
Transepithelial electrical resistance (TEER) measurement in the MucilAir™ samples after exposure to complete emissions and control air. (**A**) One-day exposure; a significant drop was observed for both exposed and control samples compared to T0. (**B**) Five-day exposure; a decrease of TEER values at T3–T5 in the exposed vs. control samples was observed. At T5, a significant decrease/increase of TEER values when compared with T0 exposed/control samples was detected. Asterisks denote the significance between T0 and later time points (* *p* ≤ 0.05). Asterisks above brackets denote the significance between the respective exposed and control samples.

**Figure 2 ijms-22-00079-f002:**
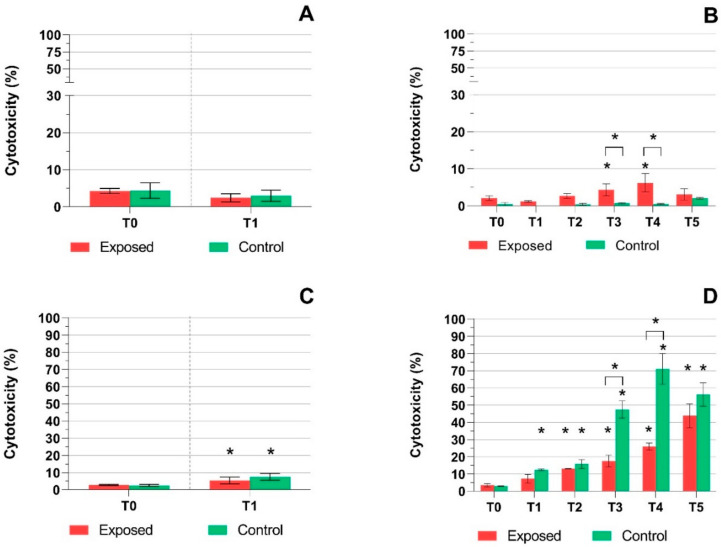
The activity of lactate dehydrogenase after exposure to complete emissions and control air. (**A**) No differences for the MucilAir™ system were detected after one-day exposure. (**B**) Five days of exposure caused an increase (* *p* ≤ 0.05) in exposed samples (T3–T4). (**C**) Elevated LDH activity in exposed and control BEAS-2B cells after one-day exposure (* *p* ≤ 0.05). (**D**) Increasing LDH activity in BEAS-2B cells after five days of treatment in both exposed and control samples (* *p* ≤ 0.05). Asterisks denote significant (* *p* ≤ 0.05) differences between T0 and later time points. Asterisks above brackets denote the significance between the respective exposed and control samples.

**Figure 3 ijms-22-00079-f003:**
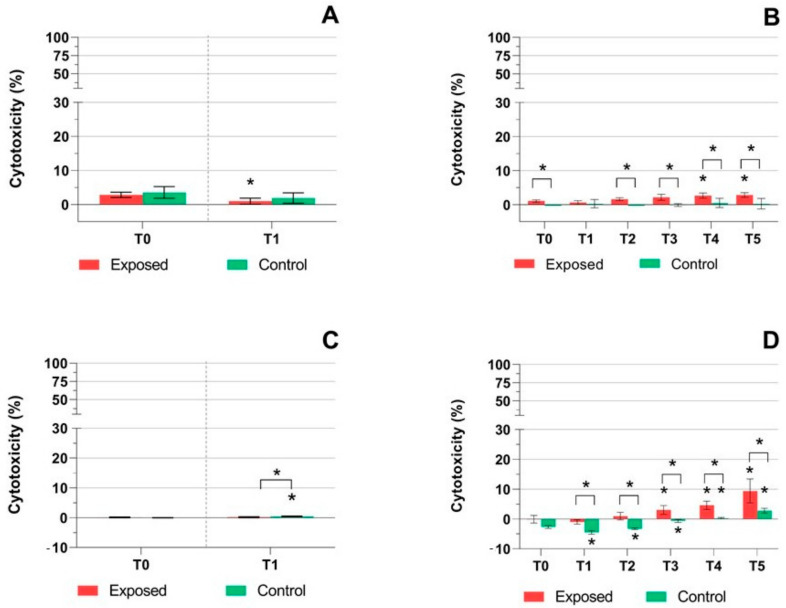
The activity of adenylate kinase after exposure to complete emissions or control air. (**A**) In MucilAir™, system activity was significantly decreased (* *p* ≤ 0.05) after one-day exposure to emissions. (**B**) After five days of exposure, the activity of adenylate kinase increased (* *p* ≤ 0.05) in the exposed samples (T0, T2–T5). (**C**) Minimal adenylate kinase activity was observed in BEAS-2B cells after one-day exposure for exposed samples. (**D**) A significant response at T3–T5 (* *p* ≤ 0.05) in BEAS-2B cells was observed after five days of treatment in exposed samples. Asterisks above brackets denote the significance between the respective exposed and control samples.

**Figure 4 ijms-22-00079-f004:**
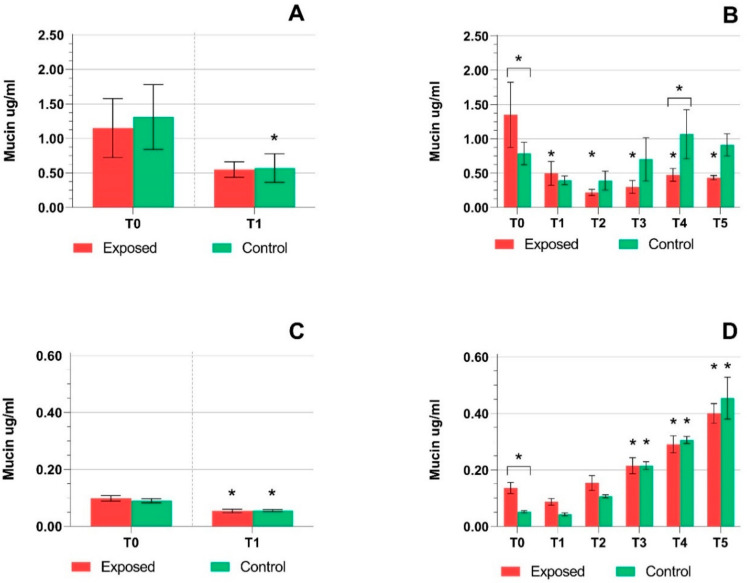
Mucin production by the MucilAir™ samples and BEAS-2B cells. (**A**) A decrease in mucin production by the MucilAir™ after one day of treatment. (**B**) A significant decrease of mucin levels after prolonged exposure of the MucilAir™ (time points T1–T5) to complete emissions and a significant difference between control and exposed samples (T0, T4). (**C**) Decreased mucin levels in BEAS-2B cells after short treatment. (**D**) Increasing mucin concentrations after five days of exposure. Asterisks denote significant (* *p* ≤ 0.05) differences between T0 and longer exposure periods. Asterisks above brackets denote the significance between the respective exposed and control samples.

**Figure 5 ijms-22-00079-f005:**
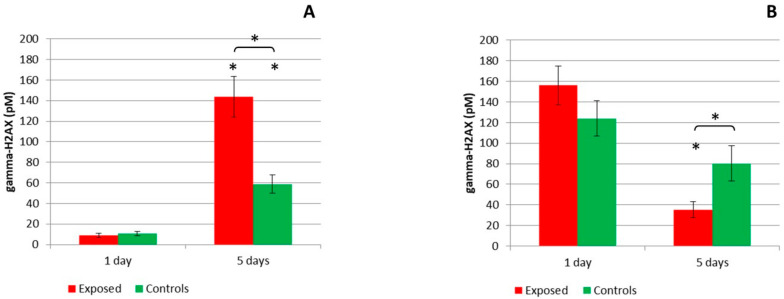
Histone H2AX phosphorylation after exposure to complete emissions and control air. (**A**) MucilAir™ system showed a significant increase (* *p* ≤ 0.05) of DNA damage after five days of exposure for both exposed and control samples compared to one-day exposure. A significant (* *p* ≤ 0.05) difference between control and exposed samples after five days of exposure was also observed. (**B**) A significant decrease (* *p* ≤ 0.05) of histone H2AX phosphorylation was detected in BEAS-2B cells after five days of exposure compared to one-day exposure. A significant (* *p* ≤ 0.05) difference between control and exposed samples was found after five days of exposure. Asterisks denote significant (* *p* ≤ 0.05) differences between one-day and five-day exposure. Asterisks above brackets denote the significance between the respective exposed and control samples.

**Figure 6 ijms-22-00079-f006:**
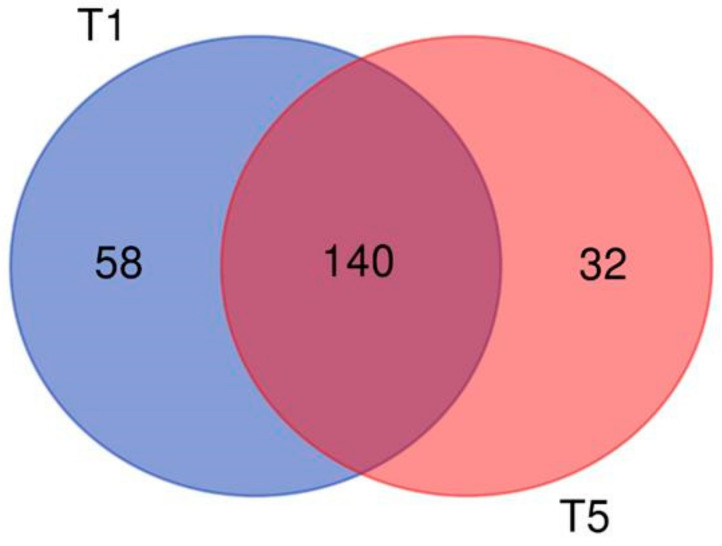
Numbers of unique and common significantly differentially expressed genes for a comparison between MucilAir™ tissues and BEAS-2B cells after one-day (T1) and five-day (T5) exposure.

**Figure 7 ijms-22-00079-f007:**
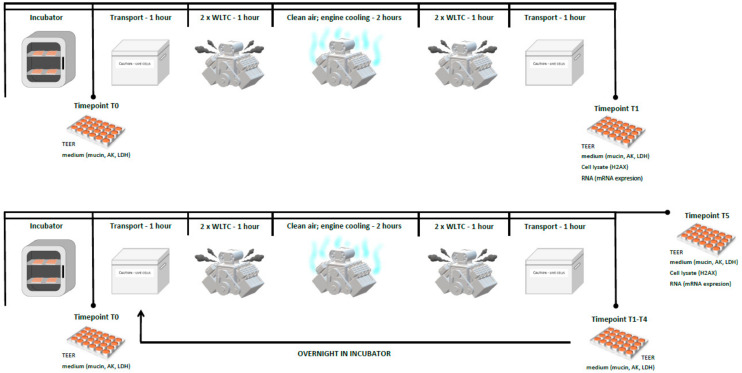
A scheme illustrating exposure of MucilAir™ and BEAS-2B cells to complete emissions. Individual time points (T0–T5) and assessed parameters are shown. Details are provided in Section 4.3. WLTC: World Harmonized Light Vehicle Test Cycle; AK: adenylate kinase; LDH: lactate dehydrogenase.

**Table 1 ijms-22-00079-t001:** The expression of genes induced by exposure to complete emissions in the MucilAir™ tissues.

Exposure Time	Gene Name	Ensemble ID	Biological Pathway	Log2 FC	Adj. *p*-Value
**Five days**	*CYP1A1*	ENSG00000140465	Cytochrome P450s and Phase I Drug Metabolism	−2.78	0.019
	*GSTA3*	ENSG00000174156	Immunotoxicity	1.46	0.019

**Table 2 ijms-22-00079-t002:** The expression of genes induced by exposure to complete emissions in BEAS-2B cells.

**A.** The expression of genes induced by exposure to complete emissions in BEAS-2B cells for one day					
**Exposure Time**	**Gene Name**	**Ensemble ID**	**Biological Pathway**	**Log2 FC**	**Adj. *p*-Value**
**One day**	*BID*	ENSG00000015475	Apoptosis	1.64	0.015
	*CASP1*	ENSG00000137752	Apoptosis	1.33	0.023
	*TNFSF10*	ENSG00000121858	Apoptosis	0.96	0.047
	*ESD*	ENSG00000139684	Cytochrome P450s and Phase I Drug Metabolism	1.93	0.008
	*MAOB*	ENSG00000069535	Cytochrome P450s and Phase I Drug Metabolism	−0.84	0.043
	*MLH1*	ENSG00000076242	DNA Damage and Repair	2.41	0.006
	*MSH2*	ENSG00000095002	DNA Damage and Repair	2.09	0.008
	*PARP1*	ENSG00000143799	DNA Damage and Repair	1.54	0.014
	*PCNA*	ENSG00000132646	DNA Damage and Repair	1.89	0.027
	*ATF4*	ENSG00000128272	Endoplasmic Reticulum Stress and Unfolded Protein Response	1.52	0.023
	*DERL1*	ENSG00000136986	Endoplasmic Reticulum Stress and Unfolded Protein Response	1.87	0.013
	*HERPUD1*	ENSG00000051108	Endoplasmic Reticulum Stress and Unfolded Protein Response	2.05	0.007
	*SERP1*	ENSG00000120742	Endoplasmic Reticulum Stress and Unfolded Protein Response	1.37	0.019
	*SYVN1*	ENSG00000162298	Endoplasmic Reticulum Stress and Unfolded Protein Response	1.91	0.014
	*ACAA2*	ENSG00000167315	Fatty Acid Metabolism	2.54	0.006
	*BDH2*	ENSG00000164039	Fatty Acid Metabolism	1.55	0.019
	*DNAJA1*	ENSG00000086061	Heat Shock Response	1.60	0.014
	*DNAJA3*	ENSG00000103423	Heat Shock Response	2.10	0.008
	*HSP90B1*	ENSG00000166598	Heat Shock Response	2.44	0.008
	*HSPA1A*	ENSG00000204389	Heat Shock Response	2.01	0.008
	*HSPA8*	ENSG00000109971	Heat Shock Response	1.96	0.008
	*HSPA9*	ENSG00000113013	Heat Shock Response	1.29	0.027
	*HSPD1*	ENSG00000144381	Heat Shock Response	1.55	0.014
	*HSPE1*	ENSG00000115541	Heat Shock Response	2.72	0.006
	*TCP1*	ENSG00000120438	Heat Shock Response	2.05	0.008
	*METAP2*	ENSG00000111142	Immunotoxicity	1.26	0.019
	*MDH1*	ENSG00000014641	Mitochondrial Energy Metabolism	1.97	0.008
	*MDH2*	ENSG00000146701	Mitochondrial Energy Metabolism	1.92	0.014
	*SDHD*	ENSG00000204370	Mitochondrial Energy Metabolism	1.90	0.008
	*SUCLA2*	ENSG00000136143	Mitochondrial Energy Metabolism	1.45	0.020
	*HOXA3*	ENSG00000105997	Necrosis	2.24	0.008
	*GPX1*	ENSG00000233276	Oxidative Stress and Antioxidant Response	1.47	0.027
	*SOD1*	ENSG00000142168	Oxidative Stress and Antioxidant Response	2.16	0.008
	*ASAH1*	ENSG00000104763	Phospholipidosis	2.36	0.006
	*MRPS18B*	ENSG00000204568	Phospholipidosis	1.93	0.010
	*ADK*	ENSG00000156110	Steatosis	1.33	0.019
	*COMT*	ENSG00000093010	Steatosis	2.08	0.008
	*HADHB*	ENSG00000138029	Steatosis	2.47	0.006
	*LY6D*	ENSG00000167656	Steatosis	2.41	0.008
**B.** The expression of genes induced by exposure to complete emissions in BEAS-2B cells for five days					
**Exposure Time**	**Gene Name**	**Ensemble ID**	**Biological Pathway**	**Log2 FC**	**Adj. *p*-Value**
**Five day**	*BIRC3*	ENSG00000023445	Apoptosis	−0.971	0.000
	*MCL1*	ENSG00000143384	Apoptosis	−0.285	0.038
	*IL6*	ENSG00000136244	Cholestasis and Immunotoxicity	−2.227	0.000
	*MDM2*	ENSG00000135679	DNA Damage and Repair	0.548	0.006
	*ACOX1*	ENSG00000161533	Fatty Acid Metabolism	0.575	0.024
	*CPT2*	ENSG00000157184	Fatty Acid Metabolism	0.472	0.038
	*IL1A*	ENSG00000115008	Immunotoxicity	−1.118	0.000
	*PTGS2*	ENSG00000073756	Immunotoxicity	−1.002	0.018
	*IDH3B*	ENSG00000101365	Mitochondrial Energy Metabolism	−0.480	0.018
	*NUDT15*	ENSG00000136159	Oxidative Stress and Antioxidant Response	−0.557	0.018
	*SLC2A3*	ENSG00000059804	Phospholipidosis	−0.989	0.028

**Table 3 ijms-22-00079-t003:** Over-represented pathways detected in BEAS-2B cells treated with complete emissions for one day and five days and in MucilAir™ when compared with BEAS-2B cells (results for individual exposure periods and regardless of the treatment).

Pathway ID	Name	*q*-Value	Deregulated Genes (N)	Genes in Pathway (N)
**BEAS-2B, 1-day exposure**				
413348	Citrate cycle	<0.001	4	14
83045	Mismatch repair	<0.001	3	23
**BEAS-2B, 5-day exposure**				
812256	TNF signaling pathway	<0.01	3	108
83105	Pathways in cancer	<0.01	4	395
**MucilAir™ vs. BEAS-2B, 1-day exposure**				
868084	Fatty acid metabolism	<0.001	8	48
814926	Carbon metabolism	<0.001	8	114
**MucilAir™ vs. BEAS-2B, 5-day exposure**				
83060	Apoptosis	<0.001	6	138
413381	beta-Oxidation	<0.001	3	12
83055	p53 signaling pathway	0.001	4	69
**MucilAir™ vs. BEAS-2B, regardless of the exposure period**				
83060	Apoptosis	<0.001	16	138
82927	Citrate cycle	<0.001	9	30
83032	Drug metabolism—cytochrome P450	<0.001	11	70

**Table 4 ijms-22-00079-t004:** Average particle concentrations and average filter loading in diluted and undiluted engine exhaust.

Exhaust	WLTC	Average Filter Loading Per Test [mg]	PM Mean [mg/m^3^]	Average Particle Number Concentrations [#/cm^3^]
**Diluted 10:1 exhaust to cells**	Cold start	0.438 ± 0.174	0.201 ± 0.089	5–8 × 10^5^
	Warm start	0.333 ± 0.099	0.148 ± 0.050	2 × 10^5^
	Average	0.385	0.175	
	**From filter** **measurements**	**From size distributions** (**EEPS, with dilution at 150 °C)**	**From size distributions** **(ELPI, no dilution at 160 °C)**	**From photoacoustic** **measurements (soot)**
**Mass concentrations in** **undiluted exhaust [mg/m^3^]**	1.75	~0.6–0.7	~0.6–0.7	~0.3

**Table 5 ijms-22-00079-t005:** Average flow rate, volume of diluted exhaust, and deposition per insert.

Exposure	Flow Rate Per Insert [cm^3^/min]	Total Volume of Diluted Exhaust Per Insert [dm^3^]	Total PM in Diluted Exhaust Per Insert [ng]	Deposition Per Insert (Estimated Deposition Rate 2%) [ng]
**One-day**	25	3	525	≈10
**Five-day**	25	15	2625	≈50

## Data Availability

The data presented in this study are available in the Appendix A.

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
