# Peer review of "Ordinary Gasoline Emissions Induce a Toxic Response in Bronchial Cells Grown at Air-Liquid Interface"

_ijms, 2020, doi:10.3390/ijms22010079_

Round 1
Reviewer 1 Report
The changes made to the manuscript are well done. The article will be of interest to the scientific community.
Author Response
We thank the reviewer for this encouraging comment.
Reviewer 2 Report
In this study, the authors explored the toxicity of ordinary gasoline exhaust in a 3D model of the human airway (MucilAirTM) and in human bronchial epithelial cells (BEAS-2B) growing at the air-liquid interface. While the findings from the current study provide some insight into the toxicity of ordinary gasoline exhaust on bronchial cells, some further information needs to be provided to improve the understanding and the rigor of the current study findings.
- Page 1, line 43: In “toxic response in MucilAirTM, In BEAS-2B cells”, the “,” should be “.”.
- Page 3, Lines 137-139: These two sentences are not clear. “We observed a significant difference (p ≤ 0.05) between the exposed and control samples at T1-T5. There were again no consistent differences between the control and exposed samples, suggesting a minor role of exposure to AK activity.”
- What kind of statistical analysis was used to obtain the difference between exposed and control groups for T3 and T4 in Fig. 1B?
- In Fig. 1 legend, the authors stated that “Asterisks denote the significance between T0 and later time points (p ≤ 0.05).” What denotes the significance between exposed and control groups? Please also clarify this issue in Figs. 2-5.
- In Fig. 2B, how to explain that exposure to complete emissions caused significant cytotoxicity at T3 and T4, but not T5?
- The experiment design seems confused. The conclusion of the present study is that the ordinary gasoline emissions induced a toxic response in MucilAirTM. In BEAS-2B cells, the authors stated that it was the incubation condition in the exposure system, rather than the effect of the complete emissions, to cause the responses in this cell line. Only two genes were dysregulated in MucilAirTM (Table 1). However, the authors used Table 2A, Table 2B, Table 3, and Fig. 6 to demonstrate the genes dysregulated in BEAS-2B cells, which may be caused by only the incubation condition. It will be better if the authors can focus on those two genes and study further at the mRNA and protein levels, as well as the role of those two genes in the toxic effects caused by the ordinary gasoline emissions.
Author Response
In this study, the authors explored the toxicity of ordinary gasoline exhaust in a 3D model of the human airway (MucilAirTM) and in human bronchial epithelial cells (BEAS-2B) growing at the air-liquid interface. While the findings from the current study provide some insight into the toxicity of ordinary gasoline exhaust on bronchial cells, some further information needs to be provided to improve the understanding and the rigor of the current study findings.
Response: We thank the reviewer for the valuable comments that helped to improve out manuscript.
Page 1, line 43: In “toxic response in MucilAirTM, In BEAS-2B cells”, the “,” should be “.”.
Response: The sentence was corrected as indicated (line 43).
Page 3, Lines 137-139: These two sentences are not clear. “We observed a significant difference (p ≤ 0.05) between the exposed and control samples at T1-T5. There were again no consistent differences between the control and exposed samples, suggesting a minor role of exposure to AK activity.”
Response: We re-worded the first of these sentences and deleted the second one, as it is redundant in the current context of the manuscript (line 139-142).
What kind of statistical analysis was used to obtain the difference between exposed and control groups for T3 and T4 in Fig. 1B?
Response: For this comparison two-way ANOVA with Sidak's (post hoc) multiple comparison test was used. These tests (along with Student’s t-test where applicable) were applied also for other comparisons reported in the manuscript. The details are provided in Section 4.9. Statistical analysis.
In Fig. 1 legend, the authors stated that “Asterisks denote the significance between T0 and later time points (p ≤ 0.05).” What denotes the significance between exposed and control groups? Please also clarify this issue in Figs. 2-5.
Response: The difference is denoted by asterisks above brackets that identify exposed-control pairs that are compared. The legends in Fig. 1-5 were modified accordingly (line 168-168, 174, 179, 184-185, 191-192).
In Fig. 2B, how to explain that exposure to complete emissions caused significant cytotoxicity at T3 and T4, but not T5?
Response: The explanation of this result is rather difficult. As cytotoxicity was overall relatively low (maximum LDH leakage around 5%), we assume that the discrepancy between the data observed at T3, T4 and T5 probably reflects fluctuations in biological response of the system. The explanation was added to the text (line 124-125).
The experiment design seems confused. The conclusion of the present study is that the ordinary gasoline emissions induced a toxic response in MucilAirTM. In BEAS-2B cells, the authors stated that it was the incubation condition in the exposure system, rather than the effect of the complete emissions, to cause the responses in this cell line. Only two genes were dysregulated in MucilAirTM (Table 1). However, the authors used Table 2A, Table 2B, Table 3, and Fig. 6 to demonstrate the genes dysregulated in BEAS-2B cells, which may be caused by only the incubation condition. It will be better if the authors can focus on those two genes and study further at the mRNA and protein levels, as well as the role of those two genes in the toxic effects caused by the ordinary gasoline emissions.
Response: We thank the reviewer for the comment. We agree that the response of BEAS-2B cells generated a lot of data that are included in several tables, although they do not reflect the effect of emissions, but rather the cultivation conditions. Although it may seem disproportional, we designed our study with the aim to compare biological response to complete gasoline emissions in two cellular model systems (a 3D model represented by MucilAirTM and a standard cell culture represented by BEAS-2B cells), as well as to evaluate suitability of these systems for long-term exposures. We also intended this study to complement our recently published data on biological effects of exposure to emissions originating from a gasoline-ethanol blend [1]. As the overall response on gene expression level in MucilAirTM was relatively weak, although directly related to emissions exposure, we presented the data for BEAS-2B cells in more details. As for MucilAirTM exposure, the expression of CYP1A1 was studied by RT-PCR as reported in the manuscript. Expression of selected proteins was also assessed (manuscript in preparation), although CYP1A1 was not among them, as we focused on immune response-related molecules (mostly cytokines, chemokines and growth factors). Unfortunately, performing protein analyses that would expand presented data is difficult, as the amount of biological material obtained from the cell inserts is very limited (about 700 000 cells/insert) and we had to carefully select assays that will be preferably done. However, the reviewer’s suggestion is valuable and it will be used in future experiments that will be more proteomics response-oriented. We ask the reviewer for understanding of this limitation that is related to technical aspects of our exposure system. We modified the text to address this limitation of our study (line 416-420).
Reference:
- Rossner, P.; Cervena, T.; Vojtisek-Lom, M.; Vrbova, K.; Ambroz, A.; Novakova, Z.; Elzeinova, F.; Margaryan, H.; Beranek, V.; Pechout, M.; et al. The Biological Effects of Complete Gasoline Engine Emissions Exposure in a 3D Human Airway Model (MucilAirTM) and in Human Bronchial Epithelial Cells (BEAS-2B). IJMS 2019, 20, 5710, doi:10.3390/ijms20225710.
Round 2
Reviewer 2 Report
The authors answered my questions and I have no other comments.
This manuscript is a resubmission of an earlier submission. The following is a list of the peer review reports and author responses from that submission.
Round 1
Reviewer 1 Report
This study explores the effect of exposure to fumes from gasoline with 5% ethanol (E5) on the growth of a normal bronchial epithelium (BE) termed MucilAir and immortalized BE grown at the air liquid interface (ALI). The effects are evaluated by assessing integrity of intercellular interactions by TEER, mucin production, cytotoxicity and DNA damage and correlates these findings with changes in gene expression. The study is essentially a repeat of a similar study published by the team that studied E20 (20% ethanol) gas. The current study demonstrates more dramatic effects and thus is a valuable extension of the previous work.
The TEER findings show significant decreases at days 1 - 5 in both treated and untreated MucilAir. These unanticipated changes have been reported elsewhere and associated with loss of epithelium in the ALI system. Given that some similarly unexpected findings (see below) are encountered, presentation of histologic features of the ALI cultures should be included and correlation with cellularity should be evaluated.
The data in the respect to the cytotoxicity analysis are counter to expected results for the LDH. The control cells show a much higher degree of LDH increase. This also raises some concerns regarding the technique that could be assessed by histologic and other methods. Other potential explanations for these results are appropriately discussed in the discussion. Conversely the data for AK are what would be expected and support the hypothesis that gas emissions are toxic to bronchial epithelial cells. The control cultures show a decrease in AK activity that gradually increases with time whereas the treated cultures show a significant increase over time with consistently significant changes in comparison to the controls. The authors conclude for both sets of data that there is not a meaningful difference whereas, there appears to be an important difference in respect to AK activity. These findings should be more strongly described and discussed
The gH2AX results are intriguing. Although considered unexpected by the authors, they may provide even more insight to the differences between the normal MucilAir ALI and the ALI using the immortalized BEAS-2B. Inhibition of factors such as PLK1 that can induce DNA damage responses such as error prone non-homologous end joining that leads to increased gH2AX foci and overactivity of these pathways as has been seen in radiation treated lung cancer cells (PMID: 29040814). Thus, decreased gH2AX foci in the BEAS-2B (but not MucilAir) may be reflective of the difference between the two ALI models. The authors address these differences in respect to both DNA damage findings and gene expression differences. However, this aspect of the data should be more fully explored. A more in depth GSEA or pathway analysis of the gene expression data regarding the differences between these two ALI models would be desirable. Given the use of a targeted gene expression panel, this may not produce many findings and likely will not be fully representative. The limitations of the targeted vs. exome wide gene expression analysis are appropriately acknowledged in the discussion but do represent a limitation to the data.
Author Response
REVIEWER 1
This study explores the effect of exposure to fumes from gasoline with 5% ethanol (E5) on the growth of a normal bronchial epithelium (BE) termed MucilAir and immortalized BE grown at the air liquid interface (ALI). The effects are evaluated by assessing integrity of intercellular interactions by TEER, mucin production, cytotoxicity and DNA damage and correlates these findings with changes in gene expression. The study is essentially a repeat of a similar study published by the team that studied E20 (20% ethanol) gas. The current study demonstrates more dramatic effects and thus is a valuable extension of the previous work.
The TEER findings show significant decreases at days 1 - 5 in both treated and untreated MucilAir. These unanticipated changes have been reported elsewhere and associated with loss of epithelium in the ALI system. Given that some similarly unexpected findings (see below) are encountered, presentation of histologic features of the ALI cultures should be included and correlation with cellularity should be evaluated.
Response: We thank the Reviewer for this comment. Investigation of the histologic features of the ALI cultures could indeed bring interesting observations. Unfortunately, as these methods are destructive, and our system allowed us to expose a limited number of cell samples, we were not able to obtain such results due to the lack of biological material. To address possible deleterious effects of exposure on the ALI systems, we employed non-destructive methods instead to check the overall quality of the cell cultures. The samples were observed twice a day under the microscope, to check if there are any visible changes on surface and TEER measurement was repeatedly conducted. Although TEER values in T1-T5 dropped below the T0 values, tight junctions were most likely not significantly affected by the treatment, as TEER remained well above the 200 Ω*cm2 limit at all time points and there were no visible signs of damage. This fact has been mentioned in the Discussion (line 308-310).
The data in the respect to the cytotoxicity analysis are counter to expected results for the LDH. The control cells show a much higher degree of LDH increase. This also raises some concerns regarding the technique that could be assessed by histologic and other methods. Other potential explanations for these results are appropriately discussed in the discussion. Conversely the data for AK are what would be expected and support the hypothesis that gas emissions are toxic to bronchial epithelial cells. The control cultures show a decrease in AK activity that gradually increases with time whereas the treated cultures show a significant increase over time with consistently significant changes in comparison to the controls. The authors conclude for both sets of data that there is not a meaningful difference whereas, there appears to be an important difference in respect to AK activity. These findings should be more strongly described and discussed.
Response: The more significant toxic effects observed in the cells exposed to the control air than to gasoline exhaust is linked to a brief disturbance of control air distribution that occurred between T0 and T1 time points (explained in Discussion). However, in our study, this effect was transient. If we take a closer look at the cytotoxicity results, for adenylate kinase we observed a weak response (less than 10% cytotoxicity) that probably has no biological significance, and thus we did not make any conclusions even though the trends look interesting. In the manuscript, we modified figure legends (Fig. 1-4) to explain the results for the control samples. These effects are further mentioned in Discussion, line 302-308.
The gH2AX results are intriguing. Although considered unexpected by the authors, they may provide even more insight to the differences between the normal MucilAir ALI and the ALI using the immortalized BEAS-2B. Inhibition of factors such as PLK1 that can induce DNA damage responses such as error prone non-homologous end joining that leads to increased gH2AX foci and overactivity of these pathways as has been seen in radiation treated lung cancer cells (PMID: 29040814). Thus, decreased gH2AX foci in the BEAS-2B (but not MucilAir) may be reflective of the difference between the two ALI models.
Response: We agree with the Reviewer that differences between the models may be reflected also in different y-H2AX values. We added a comment to Discussion (line 350-351). As previously mentioned [1],[2], MucilAirTM was more suitable for long-term ALI exposures than BEAS-2B cells cultivated at the ALI. In our pilot experiments, we compared the control BEAS-2B cells cultivated in immerse conditions and the same cells exposed to the control air in cell inserts. The cells exposed to the airflow had 6-times higher values of y-H2AX (comparable to control samples used in this study) than the cells cultivated in immerse conditions.
The authors address these differences in respect to both DNA damage findings and gene expression differences. However, this aspect of the data should be more fully explored. A more in depth GSEA or pathway analysis of the gene expression data regarding the differences between these two ALI models would be desirable. Given the use of a targeted gene expression panel, this may not produce many findings and likely will not be fully representative. The limitations of the targeted vs. exome wide gene expression analysis are appropriately acknowledged in the discussion but do represent a limitation to the data.
Response: As requested by the Reviewer, we conducted a pathway analysis in which we investigated pathway deregulation in BEAS-2B cells exposed for 1 day and 5 days to engine emissions in comparison with the control samples, and in MucilAirTM when compared with BEAS-2B cells. The results are summarized in Table 3 and discussed in the manuscript. As mentioned by the Reviewer, the number of deregulated pathways is rather limited due to the fact that a targeted gene expression panel was used. Despite this fact, some significant differences between the compared groups were noted. They are described in the text (Results, line 259-269, Table 3; Discussion, line 422-429).
Reviewer 2 Report
In this study, the authors determined the toxicity of ordinary gasoline exhaust generated by a direct injection spark ignition engine in a 3D model of the human airway (MucilAirTM) and in human bronchial epithelial cells (BEAS-2B) growing at the air-liquid interface. The authors concluded that the ordinary gasoline emissions induced a toxic response in these in vitro models. Although it is interesting to study the toxic effects of ordinary gasoline exhaust in a setting that mimics the real-world exposure, there are several issues that need to be resolved.
- The authors concluded that the ordinary gasoline emissions induced a toxic response in the bronchial cells. However, the data provided in the manuscript seem not support this conclusion; most results in the manuscript demonstrated that there were much greater toxic effects on the cells exposed to the control air than to the ordinary gasoline exhaust. For example, in Fig. 2B and D, Fig. 3B, Fig. 4B, Fig. 5, et al, it seems that control air (control group) caused a greater toxicity in the cells than ordinary gasoline emissions (exposed group). Why?
- In Figs. 1-4, what is the difference between the one-day treatment and the T1 in the 5-day treatment?
- Fig. 5: Why did not the authors detect the baseline of y-H2AX (T0) in two systems, which was without any treatment?
- For the gene expression difference, it is better to further confirm them, for example, by real-time PCR.
Author Response
REVIEWER 2
In this study, the authors determined the toxicity of ordinary gasoline exhaust generated by a direct injection spark ignition engine in a 3D model of the human airway (MucilAirTM) and in human bronchial epithelial cells (BEAS-2B) growing at the air-liquid interface. The authors concluded that the ordinary gasoline emissions induced a toxic response in these in vitro models. Although it is interesting to study the toxic effects of ordinary gasoline exhaust in a setting that mimics the real-world exposure, there are several issues that need to be resolved.
- The authors concluded that the ordinary gasoline emissions induced a toxic response in the bronchial cells. However, the data provided in the manuscript seem not support this conclusion; most results in the manuscript demonstrated that there were much greater toxic effects on the cells exposed to the control air than to the ordinary gasoline exhaust. For example, in Fig. 2B and D, Fig. 3B, Fig. 4B, Fig. 5, et al, it seems that control air (control group) caused a greater toxicity in the cells than ordinary gasoline emissions (exposed group). Why?
Response: We thank the Reviewer for the comment. We are aware of this seemingly unexpected situation and mention it in the manuscript. The greater toxic effects in the control cells shown in Fig. 2B, Fig. 3B and Fig. 4B are linked to a brief disturbance of control air distribution that occurred between T0 and T1 time points (explained in Discussion). However, in our study, this effect was transient, with a significant difference between the exposed and control samples limited to T1 (for TEER), T1-T2 (for LDH and mucin detection) and T1-T4 (for adenylate kinase). In addition, although TEER values in T1-T5 dropped below the T0 values, tight junctions were most likely not significantly affected by the treatment, as TEER remained well above the 200 Ω*cm2 limit at all time points and there were no visible signs of damage. If we take a close look at the cytotoxicity results, we observed less than 20% cytotoxicity at T1 for LDH and less than 10% cytotoxicity at T2 for adenylate kinase. We have previously observed a similar temporary increase in the levels of cytotoxicity markers in early exposure time points, from which the cells recovered during further incubation [3]. Furthermore, we compared gene expression data from the affected controls with another control sample where there was no disturbance at any point, and we did not see any significant differences suggesting that the control group completely recovered from air-flow disturbance. The explanation is included in Figures descriptions (Fig.1-4) and in Discussion, line 302-308..
In Fig. 2D, we observed significantly higher cytotoxicity in the control group for BEAS-2B cells at T3 and T4. Unlike the MucilAirTM, BEAS-2B cells continue growing in the cell inserts and in our previous study [1], we concluded that the prolonged cultivation for 5-days may affect viability of BEAS-2B cells. We explain the higher cytotoxicity in the control group at later time-points by continuous cell proliferation and cell death due to the unfavourable conditions.
- In Figs. 1-4, what is the difference between the one-day treatment and the T1 in the 5-day treatment?
Response: One-day and 5-days treatments were conducted as separate experiments on different days. For this reason, we included separate control samples to each experiment to ensure validity. As mentioned above, the difference between one-day and 5-day control samples at T1 is linked to a brief disturbance of control air distribution that occurred between T0 and T1 time points (5-days exposure).
- 5: Why did not the authors detect the baseline of y-H2AX (T0) in two systems, which was without any treatment?
Response: We thank the Reviewer for this valuable comment. Indeed, in this study we did not assess y-H2AX levels in untreated cells, as our main objective was to detect the differences between samples exposed to engine emissions and the controls exposed to clean air in the environment of exposure chamber that in general is not favourable for cell cultivation. As we were aware of the unfavourable conditions, especially for BEAS-2B cells which are typically cultivated in immerse culture with no airflow, we concentrated on the effects of the exposure chamber itself. Thus, we conducted measurement on the cells which have been already exposed to the airflow (T1 and T5).
However, in our pilot experiments, we compared the control cells cultivated in immerse conditions and the cells exposed to the control air in cell inserts (both BEAS-2B cells). The cells exposed to the airflow had 6-times higher values of y-H2AX than the cells cultivated in immerse conditions that had barely detectable values of this parameter.
- For the gene expression difference, it is better to further confirm them, for example, by real-time PCR.
Response: As requested, we performed verification of gene expression data using qRT-PCR of selected genes that were significantly deregulated in MucilAirTM and BEAS-2B cells. Overall, we observed a significant correlation between the data (log2FC) generated by mRNA sequencing and qRT-PCR. Details are provided in Results (line 277-285), Materials and Methods (line 588-602) and Supplementary File 2.
References:
- Rossner, P.; Cervena, T.; Vojtisek-Lom, M.; Vrbova, K.; Ambroz, A.; Novakova, Z.; Elzeinova, F.; Margaryan, H.; Beranek, V.; Pechout, M.; Macoun, D.; Klema, J.; Rossnerova, A.; Ciganek, M.; Topinka, J. The Biological Effects of Complete Gasoline Engine Emissions Exposure in a 3D Human Airway Model (MucilAirTM) and in Human Bronchial Epithelial Cells (BEAS-2B). Int. J. Mol. Sci. 2019, 20 (22), 0–22. https://doi.org/10.3390/ijms20225710.
- Cervena, T.; Vrbova, K.; Rossnerova, A.; Topinka, J.; Rossner, P. Short-Term and Long-Term Exposure of the MucilAirTM Model to Polycyclic Aromatic Hydrocarbons. Altern. Lab. Anim. 2019, 47 (1), 9–18. https://doi.org/10.1177/0261192919841484.
- Vojtisek-Lom, M.; Pechout, M.; Macoun, D.; Rameswaran, R.; Praharaj, K. K.; Cervena, T.; Topinka, J.; Rossner, P. Assessing Exhaust Toxicity with Biological Detector: Configuration of Portable Air-Liquid Interface Human Lung Cell Model Exposure System, Sampling Train and Test Conditions. In SAE Technical Paper; 2019. https://doi.org/10.4271/2019-24-0050.
Round 2
Reviewer 1 Report
Although technical limitations prevented addressing some of the initial concerns, the overall changes add informative content.
Reviewer 2 Report
The authors concluded that the ordinary gasoline emissions induced a toxic response in the bronchial cells. However, there is no enough data to support this conclusion. Moreover, some data showed that there were much greater toxic effects in the control group than in the exposed group. For example, in Fig. 5, the results showed that there were more DNA damage (high yH2AX level) in the control group than in the exposed group. And such kinds of results were also observed in some panels of Figs. 2-4.